# DFacTo: Distributed Factorization of Tensors

**Joon Hee Choi**
Electrical and Computer Engineering
Purdue University
West Lafayette IN 47907
choi240@purdue.edu

**S. V. N. Vishwanathan**
Statistics and Computer Science
Purdue University
West Lafayette IN 47907
vishy@stat.purdue.edu

## Abstract

We present a technique for significantly speeding up Alternating Least Squares (ALS) and Gradient Descent (GD), two widely used algorithms for tensor factorization. By exploiting properties of the Khatri-Rao product, we show how to efficiently address a computationally challenging sub-step of both algorithms. Our algorithm, DFacTo, only requires two sparse matrix-vector products and is easy to parallelize. DFacTo is not only scalable but also on average 4 to 10 times faster than competing algorithms on a variety of datasets. For instance, DFacTo only takes 480 seconds on 4 machines to perform one iteration of the ALS algorithm and 1,143 seconds to perform one iteration of the GD algorithm on a 6.5 million $\times$ 2.5 million $\times$ 1.5 million dimensional tensor with **1.2 billion** non-zero entries.

## 1 Introduction

Tensor data appears naturally in a number of applications [1, 2]. For instance, consider a social network evolving over time. One can form a users $\times$ users $\times$ time tensor which contains snapshots of interactions between members of the social network [3]. As another example consider an online store such as Amazon.com where users routinely review various products. One can form a users $\times$ items $\times$ words tensor from the review text [4]. Similarly a tensor can be formed by considering the various contexts in which a user has interacted with an item [5]. Finally, consider data collected by the Never Ending Language Learner from the Read the Web project which contains triples of noun phrases and the context in which they occur, such as, ("George Harrison", "plays", "guitars") [6].

While matrix factorization and matrix completion have become standard tools that are routinely used by practitioners, unfortunately, the same cannot be said about tensor factorization. The reasons are not very hard to see: There are two popular algorithms for tensor factorization namely Alternating Least Squares (ALS) (Appendix B), and Gradient Descent (GD) (Appendix C). The key step in both algorithms is to multiply a matricized tensor and a Khatri-Rao product of two matrices (line 4 of Algorithm 2 and line 4 of Algorithm 3). However, this process leads to a computationally-challenging, intermediate data explosion problem. This problem is exacerbated when the dimensions of tensor we need to factorize are very large (of the order of hundreds of thousands or millions), or when sparse tensors contain millions to billions of non-zero entries. For instance, a tensor we formed using review text from Amazon.com has dimensions of 6.5 million $\times$ 2.5 million $\times$ 1.5 million and contains approximately **1.2 billion** non-zero entries.

Some studies have identified this intermediate data explosion problem and have suggested ways of addressing it. First, the Tensor Toolbox [7] uses the method of reducing indices of the tensor for sparse datasets and entrywise multiplication of vectors and matrices for dense datasets. However, it is not clear how to store data or how to distribute the tensor factorization computation to multiple machines (see Appendix D). That is, there is a lack of *distributable* algorithms in existing studies. Another possible strategy to solve the data explosion problem is to use GigaTensor [8]. Unfortunately, while GigaTensor does address the problem of parallel computation, it is relatively slow. To

summarize, existing algorithms for tensor factorization such as the excellent Tensor Toolbox of [7], or the Map-Reduce based GigaTensor algorithm of [8] often do not scale to large problems.

In this paper, we introduce an efficient, scalable and distributed algorithm, DFacTo, that addresses the data explosion problem. Since most large-scale real datasets are sparse, we will focus exclusively on sparse tensors. This is well justified because previous studies have shown that designing specialized algorithms for sparse tensors can yield significant speedups [7]. We show that DFacTo can be applied to both ALS and GD, and naturally lends itself to a distributed implementation. Therefore, it can be applied to massive real datasets which cannot be stored and manipulated on a single machine. For ALS, DFacTo is on average around 5 times faster than GigaTensor and around 10 times faster than the Tensor Toolbox on a variety of datasets. In the case of GD, DFacTo is on average around 4 times faster than CP-OPT [9] from the Tensor Toolbox. On the Amazon.com review dataset, DFacTo only takes 480 seconds on 4 machines to perform one iteration of ALS and 1,143 seconds to perform one iteration of GD.

As with any algorithm, there is a trade-off: DFacTo uses 3 times more memory than the Tensor Toolbox, since it needs to store 3 flattened matrices as opposed to a single tensor. However, in return, our algorithm only requires two sparse matrix-vector multiplications, making DFacTo easy to implement using any standard sparse linear algebra library. Therefore, there are two merits of using our algorithm: 1) computations are distributed in a natural way; and 2) only standard operations are required.

## 2 Notation and Preliminaries

Our notation is standard, and closely follows [2]. Also see [1]. Lower case letters such as $x$ denote scalars, bold lower case letters such as $\mathbf{x}$ denote vectors, bold upper case letters such as $\mathbf{X}$ represent matrices, and calligraphic letters such as $\mathcal{X}$ denote three-dimensional tensors.

The $i$-th element of a vector $\mathbf{x}$ is written as $x_i$. In a similar vein, the $(i,j)$-th entry of a matrix $\mathbf{X}$ is denoted as $x_{i,j}$ and the $(i,j,k)$-th entry of a tensor $\mathcal{X}$ is written as $x_{i,j,k}$. Furthermore, $\mathbf{x}_{i,:}$ (resp. $\mathbf{x}_{:,i}$) denotes the $i$-th row (resp. column) of $\mathbf{X}$. We will use $\mathbf{X}_{\Omega,:}$ (resp. $\mathbf{X}_{:,\Omega}$) to denote the sub-matrix of $\mathbf{X}$ which contains the rows (resp. columns) indexed by the set $\Omega$. For instance, if $\Omega = \{2, 4\}$, then $\mathbf{X}_{\Omega,:}$ is a matrix which contains the second and fourth rows of $\mathbf{X}$. Extending the above notation to tensors, we will write $\mathbf{X}_{i,:,:}$, $\mathbf{X}_{:,j,:}$ and $\mathbf{X}_{:,:,k}$ to respectively denote the horizontal, lateral and frontal *slices* of a third-order tensor $\mathcal{X}$. The column, row, and tube *fibers* of $\mathcal{X}$ are given by $\mathbf{x}_{:,j,k}$, $\mathbf{x}_{i,:,k}$, and $\mathbf{x}_{i,j,:}$ respectively.

Sometimes a matrix or tensor may not be fully observed. We will use $\Omega^{\mathbf{X}}$ or $\Omega^{\mathcal{X}}$ respectively to denote the set of indices corresponding to the observed (or equivalently non-zero) entries in a matrix $\mathbf{X}$ or a tensor $\mathcal{X}$. Extending this notation, $\Omega_{i,:}^{\mathbf{X}}$ (resp. $\Omega_{:,j}^{\mathbf{X}}$) denotes the set of column (resp. row) indices corresponding to the observed entries in the $i$-th row (resp. $j$-th column) of $\mathbf{X}$. We define $\Omega_{i,:,:}^{\mathcal{X}}$, $\Omega_{:,j,:}^{\mathcal{X}}$, and $\Omega_{:,:,k}^{\mathcal{X}}$ analogously as the set of indices corresponding to the observed entries of the $i$-th horizontal, $j$-th lateral, or $k$-th frontal slices of $\mathcal{X}$. Also, $nnzr(\mathbf{X})$ (resp. $nnzc(\mathbf{X})$) denotes the number of rows (resp. columns) of $\mathbf{X}$ which contain at least one non-zero element.

$\mathbf{X}^{\top}$ denotes the transpose, $\mathbf{X}^{\dagger}$ denotes the Moore-Penrose pseudo-inverse, and $\|\mathbf{X}\|$ (resp. $\|\mathcal{X}\|$) denotes the Frobenius norm of a matrix $\mathbf{X}$ (resp. tensor $\mathcal{X}$) [10]. Given a matrix $\mathbf{A} \in \mathbb{R}^{n \times m}$, the linear operator $vec(\mathbf{A})$ yields a vector $\mathbf{x} \in \mathbb{R}^{nm}$, which is obtained by stacking the columns of $\mathbf{A}$. On the other hand, given a vector $\mathbf{x} \in \mathbb{R}^{nm}$, the operator $unvec_{(n,m)}(\mathbf{x})$ yields a matrix $\mathbf{A} \in \mathbb{R}^{n \times m}$.

$\mathbf{A} \otimes \mathbf{B}$ denotes the Kronecker product, $\mathbf{A} \odot \mathbf{B}$ the Khatri-Rao product, and $\mathbf{A} * \mathbf{B}$ the Hadamard product of matrices $\mathbf{A}$ and $\mathbf{B}$. The outer product of vectors $\mathbf{a}$ and $\mathbf{b}$ is written as $\mathbf{a} \circ \mathbf{b}$ (see *e.g.*, [11]). Definitions of these standard matrix products can be found in Appendix A.

### 2.1 Flattening Tensors

Just like the $vec(\cdot)$ operator flattens a matrix, a tensor $\mathcal{X}$ may also be unfolded or flattened into a matrix in three ways namely by stacking the horizontal, lateral, and frontal slices. We use $\mathbf{X}^n$ to denote the $n$-mode flattening of a third-order tensor $\mathcal{X} \in \mathbb{R}^{I \times J \times K}$; $\mathbf{X}^1$ is of size $I \times JK$, $\mathbf{X}^2$ is of size $J \times KI$, and $\mathbf{X}^3$ is of size $K \times IJ$. The following relationships hold between the entries of $\mathcal{X}$

and its unfolded versions (see Appendix A.1 for an illustrative example):

$$x_{i,j,k} = x^1_{i,j+(k-1)J} = x^2_{j,k+(i-1)K} = x^3_{k,i+(j-1)I}. \tag{1}$$

We can view $\mathbf{X}^1$ as consisting of $K$ stacked frontal slices of $\mathcal{X}$, each of size $I \times J$. Similarly, $\mathbf{X}^2$ consists of $I$ slices of size $J \times K$ and $\mathbf{X}^3$ is made up of $J$ slices of size $K \times I$. If we use $\mathbf{X}^{n,m}$ to denote the $m$-th slice in the $n$-mode flattening of $\mathcal{X}$, then observe that the following holds:

$$x^1_{i,j+(k-1)J} = x^{1,k}_{i,j}, \quad x^2_{j,k+(i-1)K} = x^{2,i}_{j,k}, \quad x^3_{k,i+(j-1)I} = x^{3,j}_{k,i}. \tag{2}$$

One can state a relationship between the rows and columns of various flattenings of a tensor, which will be used to derive our distributed tensor factorization algorithm in Section 3. The proof of the below lemma is in Appendix A.2.

**Lemma 1** *Let $(n, n') \in \{(2,1), (3,2), (1,3)\}$, and let $\mathbf{X}^n$ and $\mathbf{X}^{n'}$ be the $n$ and $n'$-mode flattening respectively of a tensor $\mathcal{X}$. Moreover, let $\mathbf{X}^{n,m}$ be the $m$-th slice in $\mathbf{X}^n$, and $\mathbf{x}^{n'}_{m,:}$ be the $m$-th row of $\mathbf{X}^{n'}$. Then, $vec(\mathbf{X}^{n,m}) = \mathbf{x}^{n'}_{m,:}$.*

# 3 DFacTo

Recall that the main challenge of implementing ALS or GD for solving tensor factorization lies in multiplying a matricized tensor and a Khatri-Rao product of two matrices: $\mathbf{X}^1 (\mathbf{C} \odot \mathbf{B})$[1] . If $\mathbf{B}$ is of size $J \times R$ and $\mathbf{C}$ is of size $K \times R$, explicitly forming $(\mathbf{C} \odot \mathbf{B})$ requires $O(JKR)$ memory and is infeasible when $J$ and $K$ are large. This is called the intermediate data explosion problem in the literature [8]. The lemma below will be used to derive our efficient algorithm, which avoids this problem. Although the proof can be inferred using results in [2], we give an elementary proof for completeness.

**Lemma 2** *The $r$-th column of $\mathbf{X}^1 (\mathbf{C} \odot \mathbf{B})$ can be computed as*

$$\left[ \mathbf{X}^1 (\mathbf{C} \odot \mathbf{B}) \right]_{:,r} = \left[ unvec_{(K,I)} \left( \left( \mathbf{X}^2 \right)^\top \mathbf{b}_{:,r} \right) \right]^\top \mathbf{c}_{:,r} \tag{3}$$

**Proof** We need to show that

$$\left[ \mathbf{X}^1 (\mathbf{C} \odot \mathbf{B}) \right]_{:,r} = \left[ unvec_{(K,I)} \left( \left( \mathbf{X}^2 \right)^\top \mathbf{b}_{:,r} \right) \right]^\top \mathbf{c}_{:,r}$$
$$= \begin{bmatrix} \mathbf{b}^\top_{:,r} \mathbf{X}^{2,1} \mathbf{c}_{:,r} \\ \vdots \\ \mathbf{b}^\top_{:,r} \mathbf{X}^{2,I} \mathbf{c}_{:,r} \end{bmatrix}.$$

Or equivalently it suffices to show that $\left[ \mathbf{X}^1 (\mathbf{C} \odot \mathbf{B}) \right]_{i,r} = \mathbf{b}^\top_{:,r} \mathbf{X}^{2,i} \mathbf{c}_{:,r}$. Using (13)

$$vec \left( \mathbf{b}^\top_{:,r} \mathbf{X}^{2,i} \mathbf{c}_{:,r} \right) = \left( \mathbf{c}^\top_{:,r} \otimes \mathbf{b}^\top_{:,r} \right) vec \left( \mathbf{X}^{2,i} \right). \tag{4}$$

Observe that $\mathbf{b}^\top_{:,r} \mathbf{X}^{2,i} \mathbf{c}_{:,r}$ is a scalar. Moreover, using Lemma 1 we can write $vec \left( \mathbf{X}^{2,i} \right) = \mathbf{x}^1_{i,:}$. This allows us to rewrite the above equation as

$$\mathbf{b}^\top_{:,r} \mathbf{X}^{2,i} \mathbf{c}_{:,r} = \left( \mathbf{x}^1_{i,:} \right)^\top \left( \mathbf{c}_{:,r} \otimes \mathbf{b}_{:,r} \right) = \left[ \mathbf{X}^1 (\mathbf{C} \odot \mathbf{B}) \right]_{i,r},$$

which completes the proof. ∎

Unfortunately, a naive computation of $\left[ \mathbf{X}^1 (\mathbf{C} \odot \mathbf{B}) \right]_{:,r}$ by using (3) does not solve the intermediate data explosion problem. This is because $\left( \mathbf{X}^2 \right)^\top \mathbf{b}_{:,r}$ produces a $KI$ dimensional vector, which is then reshaped by the $unvec_{(K,I)}(\cdot)$ operator into a $K \times I$ matrix. However, as the next lemma asserts, only a small number of entries of $\left( \mathbf{X}^2 \right)^\top \mathbf{b}_{:,r}$ are non-zero.

For convenience, let a vector produced by $(\mathbf{X}^2)^\top \mathbf{b}_{:,r}$ be $\mathbf{v}_{:,r}$ and a matrix produced by $\left[ unvec_{(K,I)}(\mathbf{v}_{:,r}) \right]^\top$ be $\mathbf{M}^r$.

**Lemma 3** *The number of non-zeros in $\mathbf{v}_{:,r}$ is at most $nnzr((\mathbf{X}^2)^\top)$ and $nnzc(\mathbf{X}^2)$.*

**Proof** Multiplying an all-zero row in $(\mathbf{X}^2)^\top$ and $\mathbf{b}_{:,r}$ produces zero. Therefore, the number of non-zeros in $\mathbf{v}_{:,r}$ is equal to the number of rows in $(\mathbf{X}^2)^\top$ that contain at least one non-zero element. Also, by definition, $nnzr((\mathbf{X}^2)^\top)$ is equal to $nnzc(\mathbf{X}^2)$. ∎

As a consequence of the above lemma, we only need to explicitly compute the non-zero entries of $\mathbf{v}_{:,r}$. However, the problem of reshaping $\mathbf{v}_{:,r}$ via the $\left[unvec_{(K,I)}(\cdot)\right]^\top$ operator still remains. The next lemma shows how to overcome this difficulty.

**Lemma 4** *The location of the non-zero entries of $\mathbf{M}^r$ depends on $(\mathbf{X}^2)^\top$ and is independent of $\mathbf{b}_{:,r}$.*

**Proof** The product of the $(k+(i-1)K)$-th row of $(\mathbf{X}^2)^\top$ and $\mathbf{b}_{:,r}$ is the $(k+(i-1)K)$-th element of $\mathbf{v}_{:,r}$. And, this element is the $(i,k)$-th entry of $\mathbf{M}^r$ by definition of $\left[unvec_{(K,I)}(\cdot)\right]^\top$. Therefore, if all the entries in the $(k+(i-1)K)$-th row of $(\mathbf{X}^2)^\top$ are zero, then the $(i,k)$-th entry of $\mathbf{M}^r$ is zero regardless of $\mathbf{b}_{:,r}$. Consequently, the location of the non-zero entries of $\mathbf{M}^r$ is independent of $\mathbf{b}_{:,r}$, and is only determined by $(\mathbf{X}^2)^\top$. ∎

Given $\mathcal{X}$ one can compute $(\mathbf{X}^2)^\top$ to know the locations of the non-zero entries of $\mathbf{M}^r$. In other words, we can infer the non-zero pattern and therefore preallocate memory for $\mathbf{M}^r$. We will show below how this allows us to perform the $\left[unvec_{(K,I)}(\cdot)\right]^\top$ operation for free.

Recall the Compressed Sparse Row (CSR) Format, which stores a sparse matrix as three arrays namely *values*, *columns*, and *rows*. Here, *values* represents the non-zero values of the matrix; while *columns* stores the column indices of the non-zero values. Also, *rows* stores the indices of the *columns* array where each row starts. For example, if a sparse matrix $\mathbf{M}^r$ is

$$\mathbf{M}^r = \begin{bmatrix} 1 & 0 & 2 \\ 0 & 3 & 4 \end{bmatrix},$$

then the CSR of $\mathbf{M}^r$ is

$$value(\mathbf{M}^r) = [\ 1 \quad 2 \quad 3 \quad 4\ ]$$
$$col(\mathbf{M}^r) = [\ 0 \quad 2 \quad 1 \quad 2\ ]$$
$$row(\mathbf{M}^r) = [\ 0 \quad 2 \quad 4\ ].$$

Different matrices with the same sparsity pattern can be represented by simply changing the entries of the *value* array. For our particular case, what this means is that we can pre-compute $col(\mathbf{M}^r)$ and $row(\mathbf{M}^r)$ and pre-allocate $value(\mathbf{M}^r)$. By writing the non-zero entries of $\mathbf{v}_{:,r}$ into $value(\mathbf{M}^r)$ we can "reshape" $\mathbf{v}_{:,r}$ into $\mathbf{M}^r$.

Let the matrix with all-zero rows in $(\mathbf{X}^2)^\top$ removed be $(\hat{\mathbf{X}}^2)^\top$. Then, Algorithm 1 shows the DFacTo algorithm for computing $\mathbf{N} := \mathbf{X}^1\,(\mathbf{C} \odot \mathbf{B})$. Here, the input values are $(\hat{\mathbf{X}}^2)^\top$, $\mathbf{B}$, $\mathbf{C}$, and $\mathbf{M}^r$ preallocated in CSR format. By storing the results of the product of $(\hat{\mathbf{X}}^2)^\top$ and $\mathbf{b}_{:,r}$ directly into $value(\mathbf{M}^r)$, we can obtain $\mathbf{M}^r$ because $\mathbf{M}^r$ was preallocated in the CSR format. Then, the product of $\mathbf{M}^r$ and $\mathbf{c}_{:,r}$ yields the $r$-th column of $\mathbf{N}$. We obtain the output $\mathbf{N}$ by repeating these two sparse matrix-vector products $R$ times.

---

**Algorithm 1**: DFacTo algorithm for Tensor Factorization

---

**1 Input:** $(\hat{\mathbf{X}}^2)^\top$, $\mathbf{B}$, $\mathbf{C}$, $value(\mathbf{M}^r)$ $col(\mathbf{M}^r)$, $row(\mathbf{M}^r)$
**2 Output:** $\mathbf{N}$
**3 while** *r=1, 2,…, R* **do**
**4**      $value(\mathbf{M}^r) \leftarrow (\hat{\mathbf{X}}^2)^\top\,\mathbf{b}_{:,r}$
**5**      $\mathbf{n}_{:,r} \leftarrow \mathbf{M}^r\,\mathbf{c}_{:,r}$
**6 end**

---

It is immediately obvious that using the above lemmas to compute $\mathbf{N}$ requires no extra memory other than storing $\mathbf{M}^r$, which contains at most $nnzc(\mathbf{X}^2) \leq |\Omega^{\mathcal{X}}|$ non-zero entries. Therefore, we

completely avoid the intermediate data explosion problem. Moreover, the same subroutine can be used for both ALS and GD (see Appendix E for detailed pseudo-code).

## 3.1 Distributed Memory Implementation

Our algorithm is easy to parallelize using a master-slave architecture of MPI(Message Passing Interface). At every iteration, the master transmits $\mathbf{A}$, $\mathbf{B}$, and $\mathbf{C}$ to the slaves. The slaves hold a fraction of the rows of $\mathbf{X}^2$ using which a fraction of the rows of $\mathbf{N}$ is computed. By performing a synchronization step, the slaves can exchange rows of $\mathbf{N}$. In ALS, this $\mathbf{N}$ is used to compute $\mathbf{A}$ which is transmitted back to the master. Then, the master updates $\mathbf{A}$, and the iteration proceeds. In GD, the slaves transmit $\mathbf{N}$ back to the master, which computes $\nabla \mathbf{A}$. Then, the master computes the step size by a line search algorithm, updates $\mathbf{A}$, and the iteration proceeds.

## 3.2 Complexity Analysis

A naive computation of $\mathbf{N}$ requires $\left(JK + \left|\Omega^{\mathcal{X}}\right|\right) R$ flops; forming $\mathbf{C} \odot \mathbf{B}$ requires $JKR$ flops and performing the matrix-matrix multiplication $\mathbf{X}^1 \left(\mathbf{C} \odot \mathbf{B}\right)$ requires $\left|\Omega^{\mathcal{X}}\right| R$ flops. Our algorithm requires only $\left(nnzc(\mathbf{X}^2) + \left|\Omega^{\mathcal{X}}\right|\right) R$ flops; $\left|\Omega^{\mathcal{X}}\right| R$ flops for computing $\mathbf{v}_{:,r}$ and $nnzc(\mathbf{X}^2)R$ flops for computing $\mathbf{M}^r \mathbf{c}_{:,r}$. Note that, typically, $nnzc(\mathbf{X}^2) \ll$ both $JK$ and $\left|\Omega^{\mathcal{X}}\right|$ (see Table 1). In terms of memory, the naive algorithm requires $O(JKR)$ extra memory, while our algorithm only requires $nnzc(\mathbf{X}^2)$ extra space to store $\mathbf{M}^r$.

# 4 Related Work

Two papers that are most closely related to our work are the GigaTensor algorithm proposed by [8] and the Sparse Tensor Toolbox of [7]. As discussed above, both algorithms attack the problem of computing $\mathbf{N}$ efficiently. In order to compute $\mathbf{n}_{:,r}$, GigaTensor computes two intermediate matrices $\mathbf{N}_1 := \mathbf{X}^1 * \left(\mathbf{1}_I \odot \left(\mathbf{c}_{:,r} \otimes \mathbf{1}_J\right)^\top\right)$ and $\mathbf{N}_2 := bin\left(\mathbf{X}^1\right) * \left(\mathbf{1}_I \odot \left(\mathbf{1}_K \otimes \mathbf{b}_{:,r}\right)^\top\right)$. Next, $\mathbf{N}_3 := \mathbf{N}_1 * \mathbf{N}_2$ is computed, and $\mathbf{n}_{:,r}$ is obtained by computing $\mathbf{N}_3 \mathbf{1}_{JK}$. As reported in [8], GigaTensor uses $2 \left|\Omega^{\mathcal{X}}\right|$ extra storage and $5 \left|\Omega^{\mathcal{X}}\right|$ flops to compute one column of $\mathbf{N}$. The Sparse Tensor Toolbox stores a tensor as a vector of non-zero values and a matrix of corresponding indices. Entries of $\mathbf{B}$ and $\mathbf{C}$ are replicated appropriately to create intermediate vectors. A Hadamard product is computed between the non-zero entries of the matrix and intermediate vectors, and a selected set of entries are summed to form columns of $\mathbf{N}$. The algorithm uses $2 \left|\Omega^{\mathcal{X}}\right|$ extra storage and $5 \left|\Omega^{\mathcal{X}}\right|$ flops to compute one column of $\mathbf{N}$. See Appendix D for a detailed illustrative example which shows all the intermediate calculations performed by our algorithm as well as the algorithm of [8] and [7].

Also, [9] suggests the gradient-based optimization of CANDECOMP/PARAFAC (CP) using the same method as [7] to compute $\mathbf{X}^1 \left(\mathbf{C} \odot \mathbf{B}\right)$. [9] refers to this gradient-based optimization algorithm as CPOPT and the ALS algorithm of CP using the method of [7] as CPALS. Following [9], we use these names, CPALS and CPOPT.

# 5 Experimental Evaluation

Our experiments are designed to study the scaling behavior of DFacTo on both publicly available real-world datasets as well as synthetically generated data. We contrast the performance of DFacTo (ALS) with GigaTensor [8] as well as with CPALS [7], while the performance of DFacTo (GD) is compared with CPOPT [9]. We also present results to show the scaling behavior of DFacTo when data is distributed across multiple machines.

**Datasets** See Table 1 for a summary of the real-world datasets we used in our experiments. The NELL-1 and NELL-2 datasets are from [8] and consists of (noun phrase 1, context, noun phrase 2) triples from the "Read the Web" project [6]. NELL-2 is a version of NELL-1, which is obtained by removing entries whose values are below a threshold.

The Yelp Phoenix dataset is from the Yelp Data Challenge [2], while Cellartracker, Ratebeer, Beeradvocate and Amazon.com are from the Stanford Network Analysis Project (SNAP) home page. All these datasets consist of product or business reviews. We converted them into a users × items × words tensor by first splitting the text into words, removing stop words, using Porter stemming [12], and then removing user-item pairs which did not have any words associated with them. In addition, for the Amazon.com dataset we filtered words that appeard less than 5 times or in fewer than 5 documents. Note that the number of dimensions as well as the number of non-zero entries reported in Table 1 differ from those reported in [4] because of our pre-processing.

| Dataset | $I$ | $J$ | $K$ | $\Omega^{\hat{X}}$ | $nnzc(\mathbf{X}^1)$ | $nnzc(\mathbf{X}^2)$ | $nnzc(\mathbf{X}^3)$ |
|---|---|---|---|---|---|---|---|
| Yelp | 45.97K | 11.54K | 84.52K | 9.85M | 4.32M | 6.11M | 229.83K |
| Cellartracker | 36.54K | 412.36K | 163.46K | 25.02M | 19.23M | 5.88M | 1.32M |
| NELL-2 | 12.09K | 9.18K | 28.82K | 76.88M | 16.56M | 21.48M | 337.37K |
| Beeradvocate | 33.37K | 66.06K | 204.08K | 78.77M | 18.98M | 12.05M | 1.57M |
| Ratebeer | 29.07K | 110.30K | 294.04K | 77.13M | 22.40M | 7.84M | 2.85M |
| NELL-1 | 2.90M | 2.14M | 25.50M | 143.68M | 113.30M | 119.13M | 17.37M |
| Amazon | 6.64M | 2.44M | 1.68M | 1.22B | 525.25M | 389.64M | 29.91M |

Table 1: Summary statistics of the datasets used in our experiments.

We also generated the following two kinds of synthetic data for our experiments:

- the number of non-zero entries in the tensor is held fixed but we vary $I$, $J$, and $K$.
- the dimensions $I$, $J$, and $K$ are held fixed but the number of non-zeros entries varies.

To simulate power law behavior, both the above datasets were generated using the following preferential attachment model [13]: the probability that a non-zero entry is added at index $(i, j, k)$ is given by $p_i \times p_j \times p_k$, where $p_i$ (resp. $p_j$ and $p_k$) is proportional to the number of non-zero entries at index $i$ (resp. $j$ and $k$).

**Implementation and Hardware**   All experiments were conducted on a computing cluster where each node has two 2.1 GHz 12-core AMD 6172 processors with 48 GB physical memory per node. Our algorithms are implemented in C++ using the Eigen library[3] and compiled with the Intel Compiler. We downloaded Version 2.5 of the Tensor Toolbox, which is implemented in MATLAB[4]. Since open source code for GigaTensor is not freely available, we developed our own version in C++ following the description in [8]. Also, we used MPICH2[5] in order to distribute the tensor factorization computation to multiple machines. All our codes are available for download under an open source license from http://www.joonheechoi.com/research.

**Scaling on Real-World Datasets**   Both CPALS and our implementation of GigaTensor are uni-processor codes. Therefore, for this experiment we restricted ourselves to datasets which can fit on a single machine. When initialized with the same starting point, DFacTo and its competing algorithms will converge to the same solution. Therefore, we only compare the CPU time per iteration of the different algorithms. The results are summarized in Table 2. On many datasets DFacTo (ALS) is around 5 times faster than GigaTensor and 10 times faster than CPALS; the differences are more pronounced on the larger datasets. Also, DFacTo (GD) is around 4 times faster than CPOPT.

The differences in performance between DFacTo (ALS) and CPALS and between DFacTo (GD) and CPOPT can partially be explained by the fact that DFacTo (ALS, GD) is implemented in C++ while CPALS and CPOPT use MATLAB. However, it must be borne in mind that both MATLAB and our implementation use an optimized BLAS library to perform their computationally intensive numerical linear algebra operations.

Compared to the Map-Reduce version implemented in Java and used for the experiments reported in [8], our C++ implementation of GigaTensor is significantly faster and more optimized. As per [8],

[2]https://www.yelp.com/dataset_challenge/dataset
[3]http://eigen.tuxfamily.org
[4]http://www.sandia.gov/~tgkolda/TensorToolbox/
[5]http://www.mpich.org/static/downloads/

| Dataset | DFacTo (ALS) | GigaTensor | CPALS | DFacTo (GD) | CPOPT |
|---|---|---|---|---|---|
| Yelp Phoenix | 9.52 | 26.82 | 46.52 | 13.57 | 45.9 |
| Cellartracker | 23.89 | 80.65 | 118.25 | 35.82 | 130.32 |
| NELL-2 | 32.59 | 186.30 | 376.10 | 80.79 | 386.25 |
| Beeradvocate | 43.84 | 224.29 | 364.98 | 94.85 | 481.06 |
| Ratebeer | 44.20 | 240.80 | 396.63 | 87.36 | 349.18 |
| NELL-1 | 322.45 | 772.24 | - | 742.67 | - |

Table 2: Times per iteration (in seconds) of DFacTo (ALS), GigaTensor, CPALS, DFacTo (GD), and CPOPT on datasets which can fit in a single machine ($R$=10).

| | DFacTo (ALS) | | | | DFacTo (GD) | | | |
|---|---|---|---|---|---|---|---|---|
| | NELL-1 | | Amazon | | NELL-1 | | Amazon | |
| Machines | Iter. | CPU | Iter. | CPU | Iter. | CPU | Iter. | CPU |
| 1 | 322.45 | 322.45 | - | - | 742.67 | 104.23 | - | - |
| 2 | 205.07 | 167.29 | - | - | 492.38 | 55.11 | - | - |
| 4 | 141.02 | 101.58 | 480.21 | 376.71 | 322.65 | 28.55 | 1143.7 | 127.57 |
| 8 | 86.09 | 62.19 | 292.34 | 204.41 | 232.41 | 16.24 | 727.79 | 62.61 |
| 16 | 81.24 | 46.25 | 179.23 | 98.07 | 178.92 | 9.70 | 560.47 | 28.61 |
| 32 | 90.31 | 34.54 | 142.69 | 54.60 | 209.39 | 7.45 | 471.91 | 15.78 |

Table 3: Total Time and CPU time per iteration (in seconds) as a function of number of machines for the NELL-1 and Amazon datasets ($R$=10).

the Java implementation took approximately 10,000 seconds per iteration to handle a tensor with around $10^9$ non-zero entries, when using 35 machines. In contrast, the C++ version was able to handle one iteration of the ALS algorithm on the NELL-1 dataset on a single machine in 772 seconds. However, because DFacto (ALS) uses a better algorithm, it is able to handsomely outperform GigaTensor and only takes 322 seconds per iteration.

Also, the execution time of DFacTo (GD) is longer than that of DFacTo (ALS) because DFacTo (GD) spends more time on the line search algorithm to obtain an appropriate step size.

**Scaling across Machines** Our goal is to study scaling behavior of the time per iteration as datasets are distributed across different machines. Towards this end we worked with two datasets. NELL-1 is a moderate-size dataset which our algorithm can handle on a single machine, while Amazon is a large dataset which does not fit on a single machine. Table 3 shows that the iteration time decreases as the number of machines increases on the NELL-1 and Amazon datasets. While the decrease in iteration time is not completely linear, the computation time excluding both synchronization and line search time decreases linearly. The Y-axis in Figure 1 indicates $T_4/T_n$ where $T_n$ is the single iteration time with $n$ machines on the Amazon dataset.

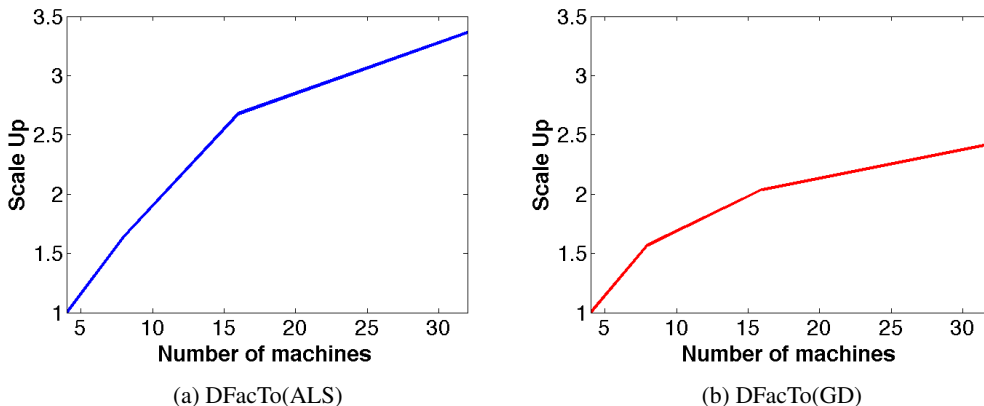

(a) DFacTo(ALS)           (b) DFacTo(GD)

Figure 1: The scalability of DFacTo with respect to the number of machines on the Amazon dataset

**Synthetic Data Experiments**  We perform two experiments with synthetically generated tensor data. In the first experiment we fix the number of non-zero entries to be $10^6$ and let $I = J = K$ and vary the dimensions of the tensor. For the second experiment we fix the dimensions and let $I = J = K$ and the number of non-zero entries is set to be $2I$. The scaling behavior of the three algorithms on these two datasets is summarized in Table 4. Since we used a preferential attachment model to generate the datasets, the non-zero indices exhibit a power law behavior. Consequently, the number of columns with non-zero elements ($nnzc(\cdot)$) for $\mathbf{X}^1$, $\mathbf{X}^2$ and $\mathbf{X}^3$ is very close to the total number of non-zero entries in the tensor. Therefore, as predicted by theory, DFacTo (ALS, GD) does not enjoy significant speedups when compared to GigaTensor, CPALS and CPOPT. However, it must be noted that DFacto (ALS) is faster than either GigaTensor or CPALS in all but one case and DFacTo (GD) is faster than CPOPT in all cases. We attribute this to better memory locality which arises as a consequence of reusing the memory for $\mathbf{N}$ as discussed in Section 3.

| $I = J = K$ | Non-zeros | DFacTo (ALS) | GigaTensor | CPALS | DFacTo (GD) | CPOPT |
|---|---|---|---|---|---|---|
| $10^4$ | $10^6$ | 1.14 | 2.80 | 5.10 | 2.32 | 5.21 |
| $10^5$ | $10^6$ | 2.72 | 6.71 | 6.11 | 5.87 | 11.70 |
| $10^6$ | $10^6$ | 7.26 | 11.86 | 16.54 | 16.51 | 29.13 |
| $10^7$ | $10^6$ | 41.64 | 38.19 | 175.57 | 121.30 | 202.71 |
| $10^4$ | $2 \times 10^4$ | 0.05 | 0.09 | 0.52 | 0.09 | 0.57 |
| $10^5$ | $2 \times 10^5$ | 0.92 | 1.61 | 1.50 | 1.81 | 2.98 |
| $10^6$ | $2 \times 10^6$ | 12.06 | 22.08 | 15.84 | 21.74 | 26.04 |
| $10^7$ | $2 \times 10^7$ | 144.48 | 251.89 | 214.37 | 275.19 | 324.2 |

Table 4: Time per iteration (in seconds) on synthetic datasets (non-zeros = $10^6$ or $2I$, R=10)

**Rank Variation Experiments**  Table 5 shows the time per iteration on various ranks ($R$) with the NELL-2 dataset. We see that the computation time of our algorithm increases linerally in $R$ like the time complexity analyzed in Section 3.2.

| $R$ | 5 | 10 | 20 | 50 | 100 | 200 | 500 |
|---|---|---|---|---|---|---|---|
| NELL-2 | 15.84 | 31.92 | 58.71 | 141.43 | 298.89 | 574.63 | 1498.68 |

Table 5: Time per iteration (in seconds) on various $R$

## 6 Discussion and Conclusion

We presented a technique for significantly speeding up the Alternating Least Squares (ALS) and the Gradient Descent (GD) algorithm for tensor factorization by exploiting properties of the Khatri-Rao product. Not only is our algorithm, DFacto, computationally attractive, but it is also more memory efficient compared to existing algorithms. Furthermore, we presented a strategy for distributing the computations across multiple machines.

We hope that the availability of a scalable tensor factorization algorithm will enable practitioners to work on more challenging tensor datasets, and therefore lead to advances in the analysis and understanding of tensor data. Towards this end we intend to make our code freely available for download under a permissive open source license.

Although we mainly focused on tensor factorization using ALS and GD, it is worth noting that one can extend the basic ideas behind DFacTo to other related problems such as joint matrix completion and tensor factorization. We present such a model in Appendix F. In fact, we believe that this joint matrix completion and tensor factorization model by itself is somewhat new and interesting in its own right, despite its resemblance to other joint models including tensor factorization such as [14]. In our joint model, we are given a user × item ratings matrix $\mathbf{Y}$, and some side information such as a user × item × words tensor $\mathcal{X}$. Preliminary experimental results suggest that jointly factorizing $Y$ and $\mathcal{X}$ outperforms vanilla matrix completion. Please see Appendix F for details of the algorithm and some experimental results.

## Footnotes

[1]We mainly concentrate on the update to $\mathbf{A}$ since the updates to $\mathbf{B}$ and $\mathbf{C}$ are analogous.

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
