[Supplementary Material]

# A  Matrix Products and Related Identities

**Definition 1** *The Kronecker product $\mathbf{A} \otimes \mathbf{B} \in \mathbb{R}^{mp \times nq}$ of matrices $\mathbf{A} \in \mathbb{R}^{m \times n}$ and $\mathbf{B} \in \mathbb{R}^{p \times q}$ is defined as*

$$\mathbf{A} \otimes \mathbf{B} = \begin{bmatrix} a_{1,1}\mathbf{B} & a_{1,2}\mathbf{B} & \dots & a_{1,n}\mathbf{B} \\ \vdots & \vdots & \vdots & \vdots \\ a_{m,1}\mathbf{B} & a_{m,2}\mathbf{B} & \dots & a_{m,n}\mathbf{B} \end{bmatrix}. \tag{5}$$

**Definition 2** *The Khatri-Rao product $\mathbf{A} \odot \mathbf{B} \in \mathbb{R}^{mp \times n}$ of matrices $\mathbf{A} \in \mathbb{R}^{m \times n}$ and $\mathbf{B} \in \mathbb{R}^{p \times n}$ is given by the Kronecker product of the corresponding columns of the two matrices:*

$$\mathbf{A} \odot \mathbf{B} = \begin{bmatrix} \mathbf{a}_{:,1} \otimes \mathbf{b}_{:,1} & \mathbf{a}_{:,2} \otimes \mathbf{b}_{:,2} & \dots & \mathbf{a}_{:,n} \otimes \mathbf{b}_{:,n} \end{bmatrix}. \tag{6}$$

**Definition 3** *The Hadamard product $\mathbf{A} * \mathbf{B} \in \mathbb{R}^{n \times m}$ of two conforming matrices $\mathbf{A} \in \mathbb{R}^{n \times m}$ and $\mathbf{B} \in \mathbb{R}^{n \times m}$ is given by*

$$\mathbf{A} * \mathbf{B} = \begin{bmatrix} a_{1,1}b_{1,1} & a_{1,2}b_{1,2} & \dots & a_{1,m}b_{1,m} \\ \vdots & \vdots & \vdots & \vdots \\ a_{n,1}b_{n,1} & a_{n,2}b_{n,2} & \dots & a_{n,m}b_{n,m} \end{bmatrix} \tag{7}$$

**Definition 4** *The outer product $\mathbf{a} \circ \mathbf{b}$ of vectors $\mathbf{a} \in \mathbb{R}^m$ and $\mathbf{b} \in \mathbb{R}^n$ is given by a matrix $\mathbf{M} \in \mathbb{R}^{m \times n}$ such that*

$$m_{i,j} = a_i b_j. \tag{8}$$

*The definition can be extended to tensors by defining the outer product $\mathbf{a} \circ \mathbf{b} \circ \mathbf{c}$ of three vectors $\mathbf{a} \in \mathbb{R}^m$, $\mathbf{b} \in \mathbb{R}^n$, and $\mathbf{c} \in \mathbb{R}^p$ as a tensor $\boldsymbol{\mathcal{M}} \in \mathbb{R}^{m \times n \times p}$ with*

$$m_{i,j,k} = a_i b_j c_k. \tag{9}$$

**Definition 5** *Given a matrix $\mathbf{A} \in \mathbb{R}^{n \times m}$, the linear operator $vec(\mathbf{A})$ yields a vector $\mathbf{x} \in \mathbb{R}^{nm}$, which is obtained by stacking the columns of $\mathbf{A}$:*

$$vec(\mathbf{A}) = \mathbf{x} = \begin{bmatrix} \mathbf{a}_{:,1} \\ \mathbf{a}_{:,2} \\ \vdots \\ \mathbf{a}_{:,n} \end{bmatrix}. \tag{10}$$

*Observe that*

$$x_{i+(j-1)n} = a_{i,j}. \tag{11}$$

*On the other hand, given a vector $\mathbf{x} \in \mathbb{R}^{nm}$, the operator $unvec_{(n,m)}(\mathbf{x})$ yields a matrix $\mathbf{A} \in \mathbb{R}^{n \times m}$:*

$$unvec_{(n,m)}(\mathbf{x}) = \mathbf{A} = \begin{bmatrix} \mathbf{a}_{:,1} & \mathbf{a}_{:,2} & \dots & \mathbf{a}_{:,n} \end{bmatrix}. \tag{12}$$

The Kronecker product satisfies the following well known relationship (see *e.g.*, proposition 7.1.9 of [11]):

$$vec(\mathbf{ABC}) = \left(\mathbf{C}^\top \otimes \mathbf{A}\right) vec(\mathbf{B}). \tag{13}$$

The Khatri-Rao product satisfies (see *e.g.*, chapter 2 of [1]):

$$\left(\mathbf{A} \odot \mathbf{B}\right)^\top \left(\mathbf{A} \odot \mathbf{B}\right) = \mathbf{A}^\top \mathbf{A} * \mathbf{B}^\top \mathbf{B}. \tag{14}$$

Plugging this into the definition of the Moore-Penrose pseudo-inverse [11] immediately shows that

$$\left(\mathbf{A} \odot \mathbf{B}\right)^\dagger = \left(\mathbf{A}^\top \mathbf{A} * \mathbf{B}^\top \mathbf{B}\right)^{-1} \left(\mathbf{A} \odot \mathbf{B}\right)^\top. \tag{15}$$

### A.1 An Example of Flattening Tensors

Let $\mathcal{X}$ be a $3 \times 4 \times 3$ tensor with frontal slices

$$\begin{bmatrix} 1 & 1 & 4 & 2 \\ 3 & 4 & 5 & 3 \\ 5 & 0 & 5 & 1 \end{bmatrix} \begin{bmatrix} 4 & 5 & 5 & 1 \\ 1 & 1 & 1 & 4 \\ 1 & 1 & 0 & 3 \end{bmatrix} \begin{bmatrix} 1 & 0 & 2 & 4 \\ 4 & 1 & 5 & 1 \\ 5 & 2 & 4 & 1 \end{bmatrix}, \text{ then}$$

$$\mathbf{X}^1 = \left[ \begin{array}{cccc|cccc|cccc} 1 & 1 & 4 & 2 & 4 & 5 & 5 & 1 & 1 & 0 & 2 & 4 \\ 3 & 4 & 5 & 3 & 1 & 1 & 1 & 4 & 4 & 1 & 5 & 1 \\ 5 & 0 & 5 & 1 & 1 & 1 & 0 & 3 & 5 & 2 & 4 & 1 \end{array} \right]$$

$$\mathbf{X}^2 = \left[ \begin{array}{ccc|ccc|ccc} 1 & 4 & 1 & 3 & 1 & 4 & 5 & 1 & 5 \\ 1 & 5 & 0 & 4 & 1 & 1 & 0 & 1 & 2 \\ 4 & 5 & 2 & 5 & 1 & 5 & 5 & 0 & 4 \\ 2 & 1 & 4 & 3 & 4 & 1 & 1 & 3 & 1 \end{array} \right]$$

$$\mathbf{X}^3 = \left[ \begin{array}{ccc|ccc|ccc|ccc} 1 & 3 & 5 & 1 & 4 & 0 & 4 & 5 & 5 & 2 & 3 & 1 \\ 4 & 1 & 1 & 5 & 1 & 1 & 5 & 1 & 0 & 1 & 4 & 3 \\ 1 & 4 & 5 & 0 & 1 & 2 & 2 & 5 & 4 & 4 & 1 & 1 \end{array} \right]$$

### A.2 Proof of Lemma 1

**Proof** Using (1) and (2), we can write

$$x^3_{k,i+(j-1)I} = x^{1,k}_{i,j} = x^1_{i,j+(k-1)J}.$$

The result for $(n, n') = (1, 3)$ follows directly from (11) by letting $k = m$. For other values of $n$ and $n'$, the arguments are analogous. ∎

# B Review of ALS

In this section, we will introduce the CANDECOMP/PARAFAC(CP) decomposition model, and the ALS algorithm. The CP decomposition is a multi-way tensor factorization model. Given a tensor $\mathcal{X} \in \mathbb{R}^{I \times J \times K}$, the $R$-rank CP decomposition of $\mathcal{X}$ is given by three matrices $\mathbf{A} \in \mathbb{R}^{I \times R}$, $\mathbf{B} \in \mathbb{R}^{J \times R}$, and $\mathbf{C} \in \mathbb{R}^{K \times R}$ such that

$$\mathcal{X} \approx \sum_{r=1}^{R} \lambda_r \cdot \mathbf{a}_{:,r} \circ \mathbf{b}_{:,r} \circ \mathbf{c}_{:,r}. \tag{16}$$

Note that the columns of $\mathbf{A}$, $\mathbf{B}$, and $\mathbf{C}$ are normalized to have unit length. The CP decomposition is computed by solving

$$\min_{\hat{\mathcal{X}}} \left\| \mathcal{X} - \hat{\mathcal{X}} \right\| \quad \text{with} \quad \hat{\mathcal{X}} = \sum_{r=1}^{R} \lambda_r \cdot \mathbf{a}_{:,r} \circ \mathbf{b}_{:,r} \circ \mathbf{c}_{:,r}. \tag{17}$$

The most popular method to solve the above problem is the Alternating Least Squares (ALS) algorithm [2]. The basic idea here is to fix all the matrices except one, and solve a least squares problem. Fixing $\mathbf{B}$ and $\mathbf{C}$ and rewriting (17), this amounts to setting

$$\hat{\mathbf{A}} \leftarrow \operatorname*{argmin}_{\hat{\mathbf{A}}} \left\| \mathbf{X}^1 - \hat{\mathbf{A}} \left( \mathbf{C} \odot \mathbf{B} \right)^\top \right\| \tag{18}$$

The optimal solution of (18) can be rewritten using (15) as

$$\hat{\mathbf{A}} = \mathbf{X}^1 \left( \left( \mathbf{C} \odot \mathbf{B} \right)^\top \right)^\dagger \tag{19}$$

$$= \mathbf{X}^1 \left( \mathbf{C} \odot \mathbf{B} \right) \left( \mathbf{C}^\top \mathbf{C} * \mathbf{B}^\top \mathbf{B} \right)^{-1}. \tag{20}$$

We obtain $\mathbf{A}$ by normalizing the columns of $\hat{\mathbf{A}}$. The ALS procedure repeats analogously to find $\hat{\mathbf{B}}$ and $\hat{\mathbf{C}}$ until a stopping criterion is met. The general CP-ALS algorithm is summarized in Algorithm 2.

---

**Algorithm 2**: CP-ALS algorithm

1 **Input:** $\mathbf{X}^1, \mathbf{X}^2, \mathbf{X}^3$
2 **Initialize:** $\mathbf{A}, \mathbf{B}, \mathbf{C}$
3 **while** *stopping criterion not met* **do**
4 $\quad \mathbf{M}_1 \leftarrow \mathbf{X}^1 \left( \mathbf{C} \odot \mathbf{B} \right)$
5 $\quad \mathbf{A} \leftarrow \mathbf{M}_1 \left( \mathbf{C}^\top \mathbf{C} * \mathbf{B}^\top \mathbf{B} \right)^{-1}$
6 $\quad$ Normalize columns of $\mathbf{A}$
7 $\quad \mathbf{M}_2 \leftarrow \mathbf{X}^2 \left( \mathbf{A} \odot \mathbf{C} \right)$
8 $\quad \mathbf{B} \leftarrow \mathbf{M}_2 \left( \mathbf{A}^\top \mathbf{A} * \mathbf{C}^\top \mathbf{C} \right)^{-1}$
9 $\quad$ Normalize columns of $\mathbf{B}$
10 $\quad \mathbf{M}_3 \leftarrow \mathbf{X}^3 \left( \mathbf{B} \odot \mathbf{A} \right)$
11 $\quad \mathbf{C} \leftarrow \mathbf{M}_3 \left( \mathbf{B}^\top \mathbf{B} * \mathbf{A}^\top \mathbf{A} \right)^{-1}$
12 $\quad$ Normalize columns of $\mathbf{C}$
13 **end**

---

In tensor factorization, occasionally the problem of overfitting occurs. Thus, we add regularization terms to the objective function. Accordingly, we obtain the following new objective function:

$$\min_{\hat{\mathcal{X}}} \left\| \mathcal{X} - \hat{\mathcal{X}} \right\| + \frac{1}{2} \lambda \left( \|\mathbf{A}\|^2 + \|\mathbf{B}\|^2 + \|\mathbf{C}\|^2 \right) \quad \text{with} \quad \hat{\mathcal{X}} = \sum_{r=1}^{R} \lambda_r \cdot \mathbf{a}_{:,r} \circ \mathbf{b}_{:,r} \circ \mathbf{c}_{:,r}. \tag{21}$$

Then, the optimal solution of (21) becomes

$$\hat{\mathbf{A}} = \mathbf{X}^1 \left( \mathbf{C} \odot \mathbf{B} \right) \left( \mathbf{C}^\top \mathbf{C} * \mathbf{B}^\top \mathbf{B} + \lambda \mathbf{I} \right)^{-1}. \tag{22}$$

## C   Review of GD

In this section, we will introduce the GD algorithm using CANDECOMP/PARAFAC(CP) decomposition model introduced in Section B. This algorithm uses the same objective function as CP-ALS except for normalization. Thus, we solve

$$\min_{\hat{\mathcal{X}}} \sum_{i,j,k} \frac{1}{2} \left( x_{i,j,k} - \hat{x}_{i,j,k} \right)^2 \quad \text{s.t. } \hat{\mathcal{X}} = \sum_{r=1}^{R} \mathbf{a}_{:,r} \circ \mathbf{b}_{:,r} \circ \mathbf{c}_{:,r} \tag{23}$$

We can rewrite the equation in (23) as

$$f = \frac{1}{2} \left\| \mathbf{X}^1 - \mathbf{A} \left( \mathbf{C} \odot \mathbf{B} \right)^\top \right\|^2 . \tag{24}$$

Next, the gradient of (24) with respect to $\mathbf{A}$ can be presented as

$$\frac{\partial}{\partial \mathbf{A}} f = -\mathbf{X}^1 \left( \mathbf{C} \odot \mathbf{B} \right) + \mathbf{A} \left( \mathbf{C}^\top \mathbf{C} * \mathbf{B}^\top \mathbf{B} \right) . \tag{25}$$

In GD, the gradient of $f$ will be written as

$$\nabla f = \begin{bmatrix} vec \left( \frac{\partial}{\partial \mathbf{A}} f \right) \\ vec \left( \frac{\partial}{\partial \mathbf{B}} f \right) \\ vec \left( \frac{\partial}{\partial \mathbf{C}} f \right) \end{bmatrix} . \tag{26}$$

Then, we can compute the factor matrices $\mathbf{A}$, $\mathbf{B}$ and $\mathbf{C}$ with $\hat{f} = f - \alpha \nabla f$. The general CP-GD algorithm is summarized in Algorithm 3.

---

**Algorithm 3**: CP-OPT algorithm

---
1 **Input:** $\mathbf{X}^1, \mathbf{X}^2, \mathbf{X}^3$
2 **Initialize:** $\mathbf{A}, \mathbf{B}, \mathbf{C}$
3 **while** *stopping criterion not met* **do**
4 $\quad \mathbf{M}_1 \leftarrow \mathbf{X}^1 \left( \mathbf{C} \odot \mathbf{B} \right)$
5 $\quad \nabla \mathbf{A} \leftarrow -\mathbf{M}_1 + \mathbf{A} \left( \mathbf{C}^\top \mathbf{C} * \mathbf{B}^\top \mathbf{B} \right)$
6 $\quad \mathbf{M}_2 \leftarrow \mathbf{X}^2 \left( \mathbf{A} \odot \mathbf{C} \right)$
7 $\quad \nabla \mathbf{B} \leftarrow -\mathbf{M}_2 + \mathbf{B} \left( \mathbf{A}^\top \mathbf{A} * \mathbf{C}^\top \mathbf{C} \right)$
8 $\quad \mathbf{M}_3 \leftarrow \mathbf{X}^3 \left( \mathbf{B} \odot \mathbf{A} \right)$
9 $\quad \nabla \mathbf{C} \leftarrow -\mathbf{M}_3 + \mathbf{C} \left( \mathbf{B}^\top \mathbf{B} * \mathbf{A}^\top \mathbf{A} \right)$
10 $\quad$ Calculate Step Size $\alpha$
11 $\quad \mathbf{A} \leftarrow \mathbf{A} - \alpha \nabla \mathbf{A}$
12 $\quad \mathbf{B} \leftarrow \mathbf{B} - \alpha \nabla \mathbf{B}$
13 $\quad \mathbf{C} \leftarrow \mathbf{C} - \alpha \nabla \mathbf{C}$
14 **end**

---

We add regularization terms to the objective function to solve the problem of overfitting. The new objective function is now

$$\min_{\hat{\mathcal{X}}} \sum_{i,j,k} \frac{1}{2} \left( x_{i,j,k} - \hat{x}_{i,j,k} \right)^2 \frac{1}{2} \lambda \left( \|\mathbf{A}\|^2 + \|\mathbf{B}\|^2 + \|\mathbf{C}\|^2 \right) \quad \text{s.t. } \hat{\mathcal{X}} = \sum_{r=1}^{R} \mathbf{a}_{:,r} \circ \mathbf{b}_{:,r} \circ \mathbf{c}_{:,r} \tag{27}$$

Then, the gradient of (27) with respect to $\mathbf{A}$ becomes

$$\frac{\partial}{\partial \mathbf{A}} f = -\mathbf{X}^1 \left( \mathbf{C} \odot \mathbf{B} \right) + \mathbf{A} \left( \mathbf{C}^\top \mathbf{C} * \mathbf{B}^\top \mathbf{B} + \lambda \mathbf{I} \right) . \tag{28}$$

# D Illustrative Example

We illustrate the differences between our algorithm for computing $\mathbf{M} := \mathbf{X}^1\left(\mathbf{C} \odot \mathbf{B}\right)$ vs the algorithms proposed by [7] and [8] on the following example: Consider $\mathcal{X} \in \mathbb{R}^{2\times3\times3}$ and let

$$\mathbf{X}^1 = \begin{bmatrix} 1 & 0 & 6 & 0 & 4 & 7 & 2 & 0 & 0 \\ 0 & 0 & 0 & 3 & 0 & 8 & 0 & 5 & 9 \end{bmatrix} \quad \text{and} \quad \mathbf{X}^2 = \begin{bmatrix} 1 & 0 & 2 & 0 & 3 & 0 \\ 0 & 4 & 0 & 0 & 0 & 5 \\ 6 & 7 & 0 & 0 & 8 & 9 \end{bmatrix}.$$

Moreover, let

$$\mathbf{B} = \begin{bmatrix} 3 & 1 \\ 1 & 1 \\ 2 & 3 \end{bmatrix} \text{ and } \mathbf{C} = \begin{bmatrix} 1 & 2 \\ 2 & 1 \\ 1 & 3 \end{bmatrix}.$$

[7] propose to store the above tensor as

$$\mathbf{v}^{\mathcal{X}} = \begin{bmatrix} 1 \\ 2 \\ 3 \\ 4 \\ 5 \\ 6 \\ 7 \\ 8 \\ 9 \end{bmatrix} \text{ and } \mathbf{S}^{\mathcal{X}} = \begin{bmatrix} 0 & 0 & 0 \\ 0 & 0 & 2 \\ 1 & 0 & 1 \\ 0 & 1 & 1 \\ 1 & 1 & 2 \\ 0 & 2 & 0 \\ 0 & 2 & 1 \\ 1 & 2 & 1 \\ 1 & 2 & 2 \end{bmatrix},$$

where $\mathbf{v}^{\mathcal{X}}$ denotes the vector of non-zero entries of $\mathcal{X}$, while $\mathbf{S}^{\mathcal{X}}$ denotes the corresponding vector of indices. The algorithm proposed in Sections 3.2.4 and 3.2.7 of [7] first computes

$$\mathbf{m}_1 = \begin{bmatrix} 1 \\ 2 \\ 3 \\ 4 \\ 5 \\ 6 \\ 7 \\ 8 \\ 9 \end{bmatrix} * \begin{bmatrix} 3 \\ 3 \\ 3 \\ 1 \\ 1 \\ 2 \\ 2 \\ 2 \\ 2 \end{bmatrix} * \begin{bmatrix} 1 \\ 1 \\ 2 \\ 2 \\ 1 \\ 1 \\ 2 \\ 2 \\ 1 \end{bmatrix} = \begin{bmatrix} 3 \\ 6 \\ 18 \\ 8 \\ 5 \\ 12 \\ 28 \\ 32 \\ 18 \end{bmatrix}.$$

The above Hadamard product involves three vectors namely $\mathbf{v}_{\mathcal{X}}$, a vector formed by repeating entries of $\mathbf{B}_{:,1}$ based on $\mathbf{S}^{\mathcal{X}}_{:,2}$, and a vector formed by repeating entries of $\mathbf{C}_{:,1}$ based on $\mathbf{S}^{\mathcal{X}}_{:,3}$. Similarly, we compute the vector below but by using $\mathbf{v}^{\mathcal{X}}$ and repeated entries from $\mathbf{B}_{:,2}$ and $\mathbf{C}_{:,2}$ respectively:

$$\mathbf{m}_2 = \begin{bmatrix} 1 \\ 2 \\ 3 \\ 4 \\ 5 \\ 6 \\ 7 \\ 8 \\ 9 \end{bmatrix} * \begin{bmatrix} 1 \\ 1 \\ 1 \\ 1 \\ 1 \\ 3 \\ 3 \\ 3 \\ 3 \end{bmatrix} * \begin{bmatrix} 2 \\ 3 \\ 1 \\ 1 \\ 3 \\ 2 \\ 1 \\ 1 \\ 3 \end{bmatrix} = \begin{bmatrix} 2 \\ 6 \\ 3 \\ 4 \\ 15 \\ 36 \\ 21 \\ 24 \\ 81 \end{bmatrix}.$$

Finally, we use

$$\mathbf{S}^{\mathcal{X}}_{:,1} = \begin{bmatrix} 0 \\ 0 \\ 1 \\ 0 \\ 1 \\ 0 \\ 0 \\ 1 \\ 1 \end{bmatrix}$$

to sum the appropriate entries of $\mathbf{m}_1$ and $\mathbf{m}_2$ to form $\mathbf{M}$:

$$\mathbf{M} = \left[ \begin{array}{cc} 3+6+8+12+28 & 2+6+4+36+21 \\ 18+5+32+18 & 3+15+24+81 \end{array} \right] = \left[ \begin{array}{cc} 57 & 69 \\ 73 & 123 \end{array} \right].$$

The algorithm uses $2\left|\Omega^{\mathcal{X}}\right|$ extra storage and $5\left|\Omega^{\mathcal{X}}\right|$ flops to compute one column of $\mathbf{M}$.

On the other hand, the algorithm of [8] computes $\mathbf{M}$ as follows:

$$\mathbf{N}_1 = \mathbf{X}^1 * \left(\mathbf{1}_I \odot (\mathbf{c}_{:,0} \otimes \mathbf{1}_J)^\top\right)$$

$$= \left[ \begin{array}{ccccccccc} 1 & 0 & 6 & 0 & 4 & 7 & 2 & 0 & 0 \\ 0 & 0 & 0 & 3 & 0 & 8 & 0 & 5 & 9 \end{array} \right] * \left[ \begin{array}{ccccccccc} 1 & 1 & 1 & 2 & 2 & 2 & 1 & 1 & 1 \\ 1 & 1 & 1 & 2 & 2 & 2 & 1 & 1 & 1 \end{array} \right]$$

$$= \left[ \begin{array}{ccccccccc} 1 & 0 & 6 & 0 & 8 & 14 & 2 & 0 & 0 \\ 0 & 0 & 0 & 6 & 0 & 16 & 0 & 5 & 9 \end{array} \right].$$

Here $\mathbf{1}_n$ denotes a vector of size $n$ with all entries set to one. Similarly, if $bin\left(\mathbf{X}^1\right)$ denotes an indicator matrix for the non-zero entries of $\mathbf{X}^1$, then

$$\mathbf{N}_2 = bin\left(\mathbf{X}^1\right) * \left(\mathbf{1}_I \odot (\mathbf{1}_K \otimes \mathbf{b}_{:,0})^\top\right)$$

$$= \left[ \begin{array}{ccccccccc} 1 & 0 & 1 & 0 & 1 & 1 & 1 & 0 & 0 \\ 0 & 0 & 0 & 1 & 0 & 1 & 0 & 1 & 1 \end{array} \right] * \left[ \begin{array}{ccccccccc} 3 & 1 & 2 & 3 & 1 & 2 & 3 & 1 & 2 \\ 3 & 1 & 2 & 3 & 1 & 2 & 3 & 1 & 2 \end{array} \right]$$

$$= \left[ \begin{array}{ccccccccc} 3 & 0 & 2 & 0 & 1 & 2 & 3 & 0 & 0 \\ 0 & 0 & 0 & 3 & 0 & 2 & 0 & 1 & 2 \end{array} \right].$$

Finally we compute $\mathbf{N}_3 = \mathbf{N}_1 * \mathbf{N}_2$ via

$$\mathbf{N}_3 = \left[ \begin{array}{ccccccccc} 3 & 0 & 12 & 0 & 8 & 28 & 6 & 0 & 0 \\ 0 & 0 & 0 & 18 & 0 & 32 & 0 & 5 & 18 \end{array} \right]$$

to obtain

$$\mathbf{m}_{:,1} = \mathbf{N}_3\, \mathbf{1}_{JK} = \left[ \begin{array}{c} 57 \\ 73 \end{array} \right].$$

To compute the second column of $\mathbf{M}$ we use

$$\mathbf{N}_1 = \mathbf{X}^1 * \left(\mathbf{1}_I \odot (\mathbf{c}_{:,1} \otimes \mathbf{1}_J)^\top\right)$$

$$= \left[ \begin{array}{ccccccccc} 1 & 0 & 6 & 0 & 4 & 7 & 2 & 0 & 0 \\ 0 & 0 & 0 & 3 & 0 & 8 & 0 & 5 & 9 \end{array} \right] * \left[ \begin{array}{ccccccccc} 2 & 2 & 2 & 1 & 1 & 1 & 3 & 3 & 3 \\ 2 & 2 & 2 & 1 & 1 & 1 & 3 & 3 & 3 \end{array} \right]$$

$$= \left[ \begin{array}{ccccccccc} 2 & 0 & 12 & 0 & 4 & 7 & 6 & 0 & 0 \\ 0 & 0 & 0 & 3 & 0 & 8 & 0 & 15 & 27 \end{array} \right].$$

$$\mathbf{N}_2 = bin\left(\mathbf{X}^1\right) * \left(\mathbf{1}_I \odot (\mathbf{1}_K \otimes \mathbf{b}_{:,1})^\top\right)$$

$$= \left[ \begin{array}{ccccccccc} 1 & 0 & 1 & 0 & 1 & 1 & 1 & 0 & 0 \\ 0 & 0 & 0 & 1 & 0 & 1 & 0 & 1 & 1 \end{array} \right] * \left[ \begin{array}{ccccccccc} 1 & 1 & 3 & 1 & 1 & 3 & 1 & 1 & 3 \\ 1 & 1 & 3 & 1 & 1 & 3 & 1 & 1 & 3 \end{array} \right]$$

$$= \left[ \begin{array}{ccccccccc} 1 & 0 & 3 & 0 & 1 & 3 & 1 & 0 & 0 \\ 0 & 0 & 0 & 1 & 0 & 3 & 0 & 1 & 3 \end{array} \right].$$

Finally we compute $\mathbf{N}_3 = \mathbf{N}_1 * \mathbf{N}_2$ via

$$\mathbf{N}_3 = \left[ \begin{array}{ccccccccc} 2 & 0 & 36 & 0 & 4 & 21 & 6 & 0 & 0 \\ 0 & 0 & 0 & 3 & 0 & 24 & 0 & 15 & 81 \end{array} \right]$$

and then compute

$$\mathbf{m}_{:,1} = \mathbf{N}_3\, \mathbf{1}_{JK} = \left[ \begin{array}{c} 69 \\ 123 \end{array} \right].$$

The algorithm uses $\max\left(J + |\Omega^{\mathcal{X}}|, K + |\Omega^{\mathcal{X}}|\right)$ extra storage and $5\left|\Omega^{\mathcal{X}}\right|$ flops to compute one column of $\mathbf{M}$.

In contrast, our algorithm computes $\mathbf{M}$ as follows:

$$\mathbf{m}_{:,0} = unvec_{(2,3)}\left(\left[\begin{array}{ccc} 1 & 0 & 6 \\ 0 & 4 & 7 \\ \hline 2 & 0 & 0 \\ 0 & 0 & 0 \\ \hline 3 & 0 & 8 \\ 0 & 5 & 9 \end{array}\right]\left[\begin{array}{c} 3 \\ 1 \\ 2 \end{array}\right]\right)^{\top}\left[\begin{array}{c} 1 \\ 2 \\ 1 \end{array}\right] = \left[\begin{array}{ccc} 15 & 18 & 6 \\ 0 & 25 & 23 \end{array}\right]\left[\begin{array}{c} 1 \\ 2 \\ 1 \end{array}\right] = \left[\begin{array}{c} 57 \\ 73 \end{array}\right]$$

$$\mathbf{m}_{:,1} = unvec_{(2,3)}\left(\left[\begin{array}{ccc} 1 & 0 & 6 \\ 0 & 4 & 7 \\ \hline 2 & 0 & 0 \\ 0 & 0 & 0 \\ \hline 3 & 0 & 8 \\ 0 & 5 & 9 \end{array}\right]\left[\begin{array}{c} 1 \\ 1 \\ 3 \end{array}\right]\right)^{\top}\left[\begin{array}{c} 2 \\ 1 \\ 3 \end{array}\right] = \left[\begin{array}{ccc} 19 & 25 & 2 \\ 0 & 27 & 32 \end{array}\right]\left[\begin{array}{c} 2 \\ 1 \\ 3 \end{array}\right] = \left[\begin{array}{c} 69 \\ 123 \end{array}\right].$$

Our algorithm only requires $nnzc(\mathbf{X}^2)$ extra storage space and $2\left|\Omega^{\mathcal{X}}\right|$ flops for computing $\mathbf{M}$.

# E  The ALS and GD algorithms of the DFacTo model

The ALS and GD algorithms of the DFacTo model (Section 3) is summarized in Algorithms 4 and 5. We can solve the problem of overfitting by adding a $\lambda \mathbf{I}$ term in $\mathbf{C}^\top \mathbf{C} * \mathbf{B}^\top \mathbf{B}$, $\mathbf{A}^\top \mathbf{A} * \mathbf{C}^\top \mathbf{C}$, and $\mathbf{B}^\top \mathbf{B} * \mathbf{A}^\top \mathbf{A}$ of Algorithms 4 (lines 7, 12, 17) and 5 (lines 7, 11, 15).

---

**Algorithm 4**: DFacTo(ALS) algorithm for Tensor Factorization

---

1  **Input:** $\mathbf{X}^1, \mathbf{X}^2, \mathbf{X}^3$
2  **Initialize:** $\mathbf{A}$, $\mathbf{B}$, $\mathbf{C}$
3  **while** *stopping criterion not met* **do**
4     **while** *r=1, 2,..., R* **do**
5        $\mathbf{n}_{:,r} \leftarrow unvec_{(K,I)} \left( \left(\mathbf{X}^2\right)^\top \mathbf{b}_{:,r} \right)^\top \mathbf{c}_{:,r}$
6     **end**
7     $\mathbf{A} \leftarrow \mathbf{N} \left(\mathbf{C}^\top \mathbf{C} * \mathbf{B}^\top \mathbf{B}\right)^{-1}$
8     Normalize columns of $\mathbf{A}$
9     **while** *r=1, 2,..., R* **do**
10        $\mathbf{n}_{:,r} \leftarrow unvec_{(I,J)} \left( \left(\mathbf{X}^3\right)^\top \mathbf{c}_{:,r} \right)^\top \mathbf{a}_{:,r}$
11    **end**
12    $\mathbf{B} \leftarrow \mathbf{N} \left(\mathbf{A}^\top \mathbf{A} * \mathbf{C}^\top \mathbf{C}\right)^{-1}$
13    Normalize columns of $\mathbf{B}$
14    **while** *r=1, 2,..., Right* **do**
15        $\mathbf{n}_{:,r} \leftarrow unvec_{(J,K)} \left( \left(\mathbf{X}^1\right)^\top \mathbf{a}_{:,r} \right)^\top \mathbf{b}_{:,r}$
16    **end**
17    $\mathbf{C} \leftarrow \mathbf{N} \left(\mathbf{B}^\top \mathbf{B} * \mathbf{A}^\top \mathbf{A}\right)^{-1}$
18    Normalize columns of $\mathbf{C}$
19 **end**

---

---

**Algorithm 5**: DFacTo(GD) algorithm for Tensor Factorization

---

1  **Input:** $\mathbf{X}^1, \mathbf{X}^2, \mathbf{X}^3$
2  **Initialize:** $\mathbf{A}$, $\mathbf{B}$, $\mathbf{C}$
3  **while** *stopping criterion not met* **do**
4     **while** *r=1, 2,..., R* **do**
5        $\mathbf{n}_{:,r} \leftarrow unvec_{(K,I)}((\mathbf{X}^2)^\top \mathbf{b}_{:,r})^\top \mathbf{c}_{:,r}$
6     **end**
7     $\nabla \mathbf{A} \leftarrow \mathbf{N} + \mathbf{A} \left(\mathbf{C}^\top \mathbf{C} * \mathbf{B}^\top \mathbf{B}\right)$
8     **while** *r=1, 2,..., R* **do**
9        $\mathbf{n}_{:,r} \leftarrow unvec_{(I,J)} \left( \left(\mathbf{X}^3\right)^\top \mathbf{c}_{:,r} \right)^\top \mathbf{a}_{:,r}$
10    **end**
11    $\nabla \mathbf{B} \leftarrow \mathbf{N} + \mathbf{B} \left(\mathbf{A}^\top \mathbf{A} * \mathbf{C}^\top \mathbf{C}\right)$
12    **while** *r=1, 2,..., Right* **do**
13        $\mathbf{n}_{:,r} \leftarrow unvec_{(J,K)} \left( \left(\mathbf{X}^1\right)^\top \mathbf{a}_{:,r} \right)^\top \mathbf{b}_{:,r}$
14    **end**
15    $\nabla \mathbf{C} \leftarrow \mathbf{N} + \mathbf{C} \left(\mathbf{B}^\top \mathbf{B} * \mathbf{A}^\top \mathbf{A}\right)$
16    $\alpha \leftarrow Linesearch(\mathbf{A}, \mathbf{B}, \mathbf{C}, \nabla \mathbf{A}, \nabla \mathbf{B}, \nabla \mathbf{C})$
17    $\mathbf{A} \leftarrow \mathbf{A} - \alpha \nabla \mathbf{A}$
18    $\mathbf{B} \leftarrow \mathbf{B} - \alpha \nabla \mathbf{B}$
19    $\mathbf{C} \leftarrow \mathbf{C} - \alpha \nabla \mathbf{C}$
20 **end**

---

# F    Joint Matrix Completion and Tensor Factorization

Generally, matrix completion is used when predicting how users will rate items based on data of how these users have previously rated other items. Occasionally, however, the accuracy of prediction from matrix completion is poor because matrix completion only uses prior information on the user, item, and rating. Thus, we suggest a joint matrix completion and tensor factorization model. In this model, we add a word count tensor $\mathcal{X}$ with user-item-word dimensions to the previous rating matrix $\mathbf{Y}$. This model is similar to [14]; but instead of sharing just one dimension (item), we introduce a model that shares both the user and item dimensions. Also, while [14] applies joint tensor completion and matrix factorization, we suggest using joint matrix completion and tensor factorization.

Our joint model can be computed by solving

$$\min_{\hat{\mathcal{X}}, \hat{\mathbf{Y}}} \sum_{(i,j) \in \Omega^{\mathbf{Y}}} \frac{1}{2} \left(y_{i,j} - \hat{y}_{i,j}\right)^2 + \mu \sum_{i,j,k} \frac{1}{2} \left(x_{i,j,k} - \hat{x}_{i,j,k}\right)^2 + \lambda \frac{1}{2} \left(\|\mathbf{A}\|^2 + \|\mathbf{B}\|^2 + \|\mathbf{C}\|^2\right) \quad (29)$$

$$\text{s.t. } \hat{\mathcal{X}} = \sum_{r=1}^{R} \mathbf{a}_{:,r} \circ \mathbf{b}_{:,r} \circ \mathbf{c}_{:,r}, \hat{\mathbf{Y}} = \sum_{r=1}^{R} \mathbf{a}_{:,r} \circ \mathbf{b}_{:,r}$$

We can rewrite the equation in (29) as

$$f = \frac{1}{2} \sum_{j \in \Omega^{\mathbf{Y}}_{i,:}} \left(y_{i,j} - \mathbf{a}_{i,:} \mathbf{b}_{j,:}^{\top}\right)^2 + \frac{1}{2} \mu \sum_{j} \left(x_{i,j}^1 - \mathbf{a}_{i,:} \left(\mathbf{C} \odot \mathbf{B}\right)_{j,:}^{\top}\right)^2 + \frac{1}{2} \lambda \mathbf{a}_{i,:} \mathbf{a}_{i,:}^{\top}. \quad (30)$$

Next, the gradient of (30) with respect to $\mathbf{a}_{i,:}$ can be presented as

$$\frac{\partial}{\partial \mathbf{a}_{i,:}} f = - \left[\mathbf{y}_{i,:} \mathbf{B} + \mu \, \mathbf{x}_{i,:}^1 \left(\mathbf{C} \odot \mathbf{B}\right)\right] + \mathbf{a}_{i,:} \left[\sum_{j \in \Omega^{\mathbf{Y}}} \mathbf{b}_{j,:}^{\top} \mathbf{b}_{j,:} + \mu \, \mathbf{C}^{\top} \mathbf{C} * \mathbf{B}^{\top} \mathbf{B} + \lambda \mathbf{I}\right]. \quad (31)$$

The two optimization methods we use to solve the minimization problem in this paper are the Gradient Descent (GD) and the Alternative Least Squares (ALS).

In GD, the gradient of $f$ will be written as

$$\nabla f = \begin{bmatrix} vec\left(\frac{\partial}{\partial \mathbf{A}} f\right) \\ vec\left(\frac{\partial}{\partial \mathbf{B}} f\right) \\ vec\left(\frac{\partial}{\partial \mathbf{C}} f\right) \end{bmatrix}. \quad (32)$$

And each $vec(\cdot)$ of (32) will be computed by the gradient of $f$ in (31) that corresponds to $\mathbf{a}_{j,:}$, $\mathbf{b}_{j,:}$ and $\mathbf{c}_{k,:}$, respectively because

$$vec\left(\frac{\partial}{\partial \mathbf{A}} f\right) = \begin{bmatrix} \left(\frac{\partial}{\partial \mathbf{a}_{1,:}} f\right)^{\top} \\ \left(\frac{\partial}{\partial \mathbf{a}_{2,:}} f\right)^{\top} \\ \vdots \\ \left(\frac{\partial}{\partial \mathbf{a}_{I,:}} f\right)^{\top} \end{bmatrix}.$$

Then, we can compute the factor matrices $\mathbf{A}$, $\mathbf{B}$ and $\mathbf{C}$ with $\hat{f} = f - \alpha \nabla f$.

On the other hand, in ALS, setting (31) to zero shows that the optimal solution of (30) is given by

$$\hat{\mathbf{a}}_{i,:} = \left[\mathbf{y}_{i,:} \mathbf{B} + \mu \, \mathbf{x}_{i,:}^1 \left(\mathbf{C} \odot \mathbf{B}\right)\right] \left[\sum_{j \in \Omega^{\mathbf{Y}}} \mathbf{b}_{j,:}^{\top} \mathbf{b}_{j,:} + \mu \, \mathbf{C}^{\top} \mathbf{C} * \mathbf{B}^{\top} \mathbf{B} + \lambda \mathbf{I}\right]^{-1}.$$

In both cases, we will use DFacTo, which we suggested in Section 3, to avoid the intermediate data explosion problem of $\mathbf{X}^1 (\mathbf{C} \odot \mathbf{B})$.

### F.1 Experimental Evaluation

We evaluate the joint tensor factorization and matrix completion model on a subset of datasets from Table 1. Arguably, our experimental evaluation is very preliminary, but promising. The experimental setup is as follows: We split each dataset into train, test, and validation. We randomly select 60% of review, rating pairs and designate them as training data. We then select 20% of the remaining review, rating pairs, discard the reviews, remove users or items which do not occur in the training data, and use it for validation. A similar procedure is used to generate the test dataset. Cellartracker and RateBeer datasets contain ratings which are not in a 0 to 5 scale. For consistency, we normalize these ratings to be in 0 to 5. Our evaluation metric is the mean square error which is given by $\sum (i,j) \in \Omega^{\mathbf{Y}}(y_{i,j} - \hat{y}_{i,j})$, were $y_{i,j}$ is a test rating and $\hat{y}_{i,j}$ is the rating predicted by our model.

We train our model with $\mu \in \left\{10^2, 10^1, ..., 10^{-9}, 10^{-10}\right\}$ and $\lambda \in \{100, 10, 1, 0.1, 0.01\}$, evaluate its performance on the validation set, and pick the best model based on its mean square error. We use this model to predict on the test dataset and report average mean square error. In Tables 6 and 7, we show the MSEs from both the matrix completion and our joint model using GD and ALS. For GD, the method of backtracking line search was used.

| Dataset | Matrix Completion | | Joint (MC + TF) | | |
|---|---|---|---|---|---|
| | $\lambda$ | Test MSE | $\mu$ | $\lambda$ | Test MSE |
| Yelp Phoenix | 10 | 3.133650 | $10^{-6}$ | 0.1 | 1.481320 |
| Cellartracker | 1 | 1.506590 | $10^{-7}$ | 1 | 0.927066 |
| Beeradvocate | 1 | 0.603431 | $10^{-7}$ | 0.1 | 0.459174 |
| Ratebeer | 0.01 | 0.390188 | $10^{-9}$ | 1 | 0.389653 |

Table 6: Best Test MSE of single matrix completion and joint matrix completion and tensor factorization model after 500 iterations using Gradient Descent.

| Dataset | Matrix Completion | | Joint (MC + TF) | | |
|---|---|---|---|---|---|
| | $\lambda$ | Test MSE | $\mu$ | $\lambda$ | Test MSE |
| Yelp Phoenix | 1 | 2.904320 | 1 | 1 | 1.944050 |
| Cellartracker | 1 | 1.148010 | 100 | 0.01 | 0.363496 |
| Beeradvocate | 0.1 | 0.465695 | 10 | 0.1 | 0.373827 |
| Ratebeer | 0.1 | 0.355989 | 0.1 | 1 | 0.318692 |

Table 7: Best Test MSE of single matrix completion and joint matrix completion and tensor factorization model after 500 iterations using ALS.

The results show that our joint model produces better MSEs than matrix completion across all datasets and methods. All in all, our joint model improves the accuracy of prediction when compared to matrix completion.