[Reviews · NeurIPS 2014]

Submitted by Assigned_Reviewer_22

By exploiting the properties of the Khatri-Rao product, this paper proposes a technique to speed up Alternating Least Squares (ALS) and Gradient Descent (GD) algorithms for tensor factorization. Furthermore, parallel algorithms are also proposed for scalability. Experiments on large-scale tensors are used for evaluation.

The problem of distributed (parallel) tensor factorization is interesting and challenging. The proposed (centralized) technique for speeding up ASL and GD seems to be interesting. But the experiments can be improved to make the paper more convincing. The detailed comments are listed as follows:

1. The title (or contribution) of this paper is to design DISTRIBUTED methods for tensor factorization. However, only one short section (Section 3.1), which contains only several sentences, is used to describe the distributed algorithms. More details about the distributed framework should be provided and studied. Actually, the distributed programming models will greatly affect the scalability of machine learning algorithms. More specifically, the authors only mention MapReduce in the paper. However, it is well known that MapReduce is not a good programming model for machine learning algorithms with many iterations. Unfortunately, the algorithms proposed in this paper need many iterations.

2. In experiments, the authors only report the results about time per iteration. As mentioned above, time per iteration cannot actually reflect the efficiency of a distributed learning algorithm. The authors should report the whole time for several iterations untill convergence, which will also include the communication time and file reading/writing overhead.

3. In Section 3, the C and B are not defined although we can find them in the supplementary materials.

4. The reference style should be consistent.
Summary: The problem of distributed (parallel) tensor factorization is interesting and challenging. The proposed (centralized) technique for speeding up ASL and GD seems to be interesting. The experiments can be improved to make the paper more convincing.

Submitted by Assigned_Reviewer_33

The paper considers the problem of tensor decomposition using ALS (alternating least squares) or GD (gradient descent) techniques. The ALS technique computes each of the three factors by fixing the other two and computing the third factor using ordinary least square (OLS). The OLS solution for a I-by-J-by-K tensor X has a closed form, which has its main computation bottleneck, the product X^1 (C dot B), where C and B are K-by-R and J-by-R matrices, C dot B is the JK-by-R Khatri-Rao product, and X^1 is the stacked frontal matrices of dimension IK-by-JK. Thus naively computing X^1 (C dot B) is inefficient because of the intermediate processing dependent on the JK columns of X^1 and JK rows of C dot B.

The main contribution of the paper is transforming the product X^1 (C dot B) by computing the rth row as stacking k vectors each of them computed as b^t X^(2,i) c, where b and c are the r^th columns of B and C and X^(2,i) is the ith J-by-K face matrix of tensor X. The main advantage of this approach is that the rth row is computed by two sparse matrix-vector computation, lending the advantages in computational efficiency, memory, and distributability.
Summary: The paper addresses an important problem and comes up with an efficient practical approach which provably obtains the same solution as prior approaches. Moreover, the approach in paper is shown to 5 to 10 times faster than existing ones and seems to be efficiently distributable because it is just implemented as two sparse matrix-vector product implementations.

The presentation in the paper is quite jarring at times mostly because of lack of space and the use of appendix for giving background on ALS and GD. I recommend moving the background section in a summarized form to the main paper while moving some of the proofs in the appendix.

Detailed comments:
line 121: Shouldn’t x^n’_{m:} be written as X^n’_{m:}?
line 122: For lemma 1, an example of 3 x 3 x 3 tensor along with a diagram would help a lot.
line 174: “of b_{:r}” -> “of v_{:r}”

Submitted by Assigned_Reviewer_38

This paper addressed a computational challenge (intermediate data explosion problem) in tensor factorization, i.e., how to compute the Khatri-Rao product efficiently when the size of the tensor is big.

The technique proposed in this work is based on exploiting properties of the Khatri-Rao product, which leads to an algorithm (DFacTo) with lower computational complexity compared to previous approaches. Lemma 1-4 are not entirely new, and I don't see too much technical difficulty in deriving them. However, the author did bring them together to aid computing the Khatri-Rao product in an efficient way.

DFacTo is parallelizable as explained by the paper, however, I am skeptical its applicability in a distributed environment, as the master and slaves nodes need to frequently communicate a huge intermediate matrix (N). As shown in Figure 1, DFacTo scales sub-linearly. Nevertheless, I guess this could be a common issue for any ALS and GD based algorithms.

The overall writing is clear and grammar is good. Most other papers discussing tensor factorization usually bought with complicated notations and terminologies, while this paper did a good job in explaining and illustrating the basic notations and algorithms in the supplementary file, which I believe would be helpful to people who are relatively new to this literature.

The large-scale experiments look promising, however, further insights might be gained with more details, e.g., are the algorithms sensitive to the selection of rank R (which may also increase the size of the intermediate result)? What would happen if the some parts of the intermediate results cannot fit in the main memory? These might be some practical issues to consider when deploying DeFacTo to a distributed computing environment.
Summary: This is a well written paper. And its technical contribution, albeit not entirely innovative, could be interesting to the NIPS community.
Author Feedback
Author rebuttal: ______________________________________________________________________________________

We thank the reviewers for their comments.

* Assigned reviewer1

> 1.1. The title (or contribution) of this paper is to design DISTRIBUTED methods for tensor factorization. However, only one short section (Section 3.1), which contains only several sentences, is used to describe the distributed algorithms.

By showing that the central operation of many tensor factorization algorithms reduces to two sparse-matrix times dense vector computations, we are able to leverage many decades of work on distributed linear algebra. The key challenge, however, is to show this equivalence between tensor factorization and matrix vector product, which is the central focus of our paper.

> 1.2. the authors only mention MapReduce in the paper.

We do NOT use MapReduce. It was mentioned only twice in the paper and that too in the context of GigaTensor. Our own implementations are based on MPI, and open source code will be made freely available for download.

> However, it is well known that MapReduce is not a good programming model for machine learning algorithms with many iterations. Unfortunately, the algorithms proposed in this paper need many iterations.

We do not use MapReduce. Moreover, many successful distributed machine learning algorithms are iterative in nature. As we show in our experiments we are able to factorize tensors with 1.2 BILLION entries using our algorithms.

> 2.1. The authors should report the whole time for several iterations until convergence, which will also include the communication time and file reading/writing overhead.

Please see Table 3. We report separately the *total* time per iteration (which includes communication and computation cost) as well as the *CPU* time per iteration. Since the I/O cost is constant, and the same across all algorithms, we do not report it.

> 2.2. Try other programming models like MPI and Spark

Our algorithms use MPI. Moreover, comparing programming models is beyond the scope of the paper.

> 3. In Section 3, the C and B are not defined although we can find them in the supplementary materials.
> 4. The reference style should be consistent.

Thanks for pointing this out. Both points will be fixed in the camera ready.

* Assigned reviewer2

> The presentation in the paper is quite jarring at times mostly because of lack of space and the use of appendix for giving background on ALS and GD. I recommend moving the background section in a summarized form to the main paper while moving some of the proofs in the appendix.

Unfortunately it was a delicate balancing act to fit both background material as well as our contribution within 8 pages. However, your point is well taken and we will try to fix this in the camera ready.

> line 121: Shouldn’t x^n’_{m:} be written as X^n’_{m:}?

As per our notation defined in Section 2, each row or column of a matrix is denoted using bold lower-case letters. Thus, x is correct.

> line 122: For lemma 1, an example of 3 x 3 x 3 tensor along with a diagram would help a lot.

Excellent suggestion! We will include this in the appendix.

> line 174: “of b_{:r}” -> “of v_{:r}”

Typo will be fixed in the camera ready. Thanks!

* Assigned reviewer3

> The overall writing is clear and grammar is good. Most other papers discussing tensor factorization usually bought with complicated notations and terminologies, while this paper did a good job in explaining and illustrating the basic notations and algorithms in the supplementary file, which I believe would be helpful to people who are relatively new to this literature.

Thank you.

> I am skeptical its applicability in a distributed environment, as the master and slaves nodes need to frequently communicate a huge intermediate matrix(N).

Almost every distributed iterative machine learning algorithm requires synchronizing its parameters once per iteration. In our case, the size of the parameter matrix A = I X R = size of N. Therefore, it is not surprising that we need to synchronize N once per iteration.

No existing algorithm, that we know of, can handle datasets of the size that the distributed version of DFacto is able to handle. If you are aware of other public datasets which are larger, please let us know and we will gladly include them in our experiments.

> are the algorithms sensitive to the selection of rank R (which may also increase the size of the intermediate result)?

For all the experiments we reported in the paper we set R=10. However, the time per iteration increases linearly in R. To show this we repeated the experiments with the NELL-2 dataset, and obtained the following time per iteration with various ranks (R):

15.84 sec (R=5),
31.92 sec (R=10),
58.71 sec (R=20),
141.43 sec (R=50),
298.89 sec (R=100),
574.63 sec (R=200),
1498.68 sec (R=500).

We will include a new experimental section in the camera ready to study the effect of R.

> What would happen if the some parts of the intermediate results cannot fit in the main memory?

Note that the size of the matrix M^r is independent of R and depends only on nnzc(X^2), and it is distributed across processors. N on the other hand is a I x R matrix and grows linearly in R. Similarly, the parameters A, B, and C also grow linearly in R. We currently assume that A, B, C, and N fit in the main memory. This is not an unreasonable assumption. For instance, the largest value of I we used in the experiments is 6.5 million. To store a I x R matrix of doubles (8 bytes) with R = 10 requires only approximately 520 MB of RAM.

Since only one column of B and C are used in algorithm 1, lines 4 and 5, and one column of N is written in line 5, one can implement a sophisticated version of our algorithm which stores these matrices on disk, and loads them into main memory one column at a time.